# Abrasion Behaviors of Silica-Reinforced Solution Styrene–Butadiene Rubber Compounds Using Different Abrasion Testers

**DOI:** 10.3390/polym16142038

**Published:** 2024-07-17

**Authors:** Eunji Chae, Seong Ryong Yang, Sung-Seen Choi

**Affiliations:** 1Department of Chemistry, Sejong University, 209 Neungdong-ro, Gwangjin-gu, Seoul 05006, Republic of Korea; eunji.chae@sejong.ac.kr; 2Hankook Tire & Technology Company, 50 Yuseong-daero, Yuseong-gu, Daejeon 34127, Republic of Korea; 21100375@hankooktech.com

**Keywords:** abrasion behavior, styrene–butadiene rubber, abrasion tester, wear particle, bound rubber, crosslink density

## Abstract

Solution styrene–butadiene rubber (SSBR) is widely used to improve the properties of tire tread compounds. Tire wear particles (TWPs), which are generated on real roads as vehicles traverse, represent one of significant sources of microplastics. In this study, four SSBR compounds were prepared using two SSBRs with high styrene (STY samples) and 1,2-unit (VIN samples) contents, along with dicyclopentadiene resin. The abrasion behaviors were investigated using four different abrasion testers: cut and chip (CC), Lambourn, DIN, and laboratory abrasion tester (LAT100). The abrasion rates observed in the Lambourn and LAT100 abrasion tests were consistent with each other, but the results of CC and DIN abrasion tests differed from them. The addition of the resin improved the abrasion rate and resulted in the generation of large wear particles. The abrasion rates of STY samples in the Lambourn and LAT100 abrasion tests were lower than those of VIN samples, whereas the values in the CC and DIN abrasion tests were higher than those of VIN samples. The wear particles were predominantly larger than 1000 μm, except for the VIN sample in the DIN abrasion test. However, TWPs > 1000 μm are rarely produced on real roads. The size distributions of wear particles > 1000 μm were 74.0–99.5%, 65.9–93.4%, 7.2–95.1%, and 37.5–83.0% in the CC, Lambourn, DIN, and LAT100 abrasion tests, respectively. The size distributions of wear particles in the Lambourn and LAT100 abrasion tests were broader than those in the other tests, whereas the distributions in the CC abrasion test were narrower. The abrasion patterns and the morphologies and size distributions of wear particles generated by the four abrasion tests varied significantly, attributable to differences in the bound rubber contents, crosslink densities, and tensile properties.

## 1. Introduction

The formulation and properties of rubber compounds influence their wear properties [1,2,3,4,5]. The abrasion properties of tire treads are closely related to tire mileage and vehicle safety [6,7,8,9,10]. Research on the abrasion behavior of tire tread compounds has primarily focused on abrasion rates and patterns [11,12,13]. Tire wear particles (TWPs), which are generated through friction between the tire tread and road surface while driving, are one of the major sources of microplastics [14,15,16,17,18,19,20,21]. The morphology and size distribution of TWPs are critical for understanding the abrasion behavior and environmental impact. However, research focusing on the abrasion behaviors associated with wear particles remains limited.

Various abrasion testers, such as cut and chip (CC), Lambourn, DIN (Deutsches Institut für Normung), and LAT100 (laboratory abrasion tester), have been used to examine the wear properties of tire tread compounds [22,23,24,25,26,27,28,29,30,31,32,33,34]. The CC abrasion test, in which the abrasion specimen is pierced or cut by a sharp blade, is a simple and fast method for assessing the abrasion properties of rubber articles under harsh conditions [26,27]. The Lambourn abrasion tester simulates the deformation of a rotating wheel under a constant load, with introducing talc powder between the specimen and abrasive to prevent smearing during the test [28,29]. The DIN abrasion test is a standardized test method to characterize the abrasion resistance of rubber (ASTM D 5963) [30], with advantages of simplicity and rapid processing [31]. The LAT100 abrasion tester can simulate a wide range of test conditions, such as the slip angle, load, speed, and temperature [32,33,34]. There are a lot of various particles including wear particles of tire and road pavement and soils from the outside on real roads. Of the four abrasion tests, the Lambourn test uses mineral particles; however, it may not apply to various driving conditions.

Silica-reinforced rubber compounds are commonly used for tire treads owing to their low rolling resistance, low hysteresis, high hardness, and excellent abrasion resistance [35,36,37,38,39,40,41]. Hydrocarbon resins, such as dicyclopentadiene (DCPD), can be used to improve the properties of rubber compounds [42,43,44,45,46]. DCPD resin is a thermoplastic unsaturated hydrocarbon resin obtained by the polymerization of monomers from C5 to C9 fractions and is commonly used in tire manufacturing [47,48,49]. Notably, solution styrene–butadiene rubber (SSBR) exhibits various microstructures that improve interactions with silica. Additionally, prior studies have demonstrated that the abrasion behavior varies depending on the type of abrasion tester [22,50].

Considering these aspects, this study was aimed at examining the influence of the SSBR type and DCPD resin on the abrasion behavior of high-loaded (80 phr) silica-filled SSBR compounds. Two SSBRs with high styrene and vinyl (or 1,2-unit) contents were used to investigated the effects of the SSBR microstructures on the abrasion behavior. Four abrasion testers of CC, Lambourn, DIN, and LAT100 were used for the abrasion tests, and the results were compared in terms of the wear particles and the abrasion rates and patterns. Differences in the abrasion behaviors were explained by the bound rubber contents, crosslink densities, and tensile properties.

## 2. Materials and Methods

### 2.1. Preparation of Samples

Four SSBR rubber compounds were prepared using two SSBRs with high styrene content (SSBR1: styrene = 40 wt%, 1,2-unit = 14 wt%, and 1,4-unit = 46 wt%) and high 1,2-unit content (SSBR2: styrene = 15 wt%, 1,2-unit = 25 wt%, and 1,4-unit = 60 wt%). The detailed formulation of rubber compounds is shown in Table 1. Two compounds (STY and STYR) were made of SSBR1 and the other two (VIN and VINR) were made of SSBR2. Since SSBR1 is treated with oil of 37.5 phr because of its high molecular weight and high styrene content, the TDAE oil contents of VIN and VINR compounds are larger by 37.5 phr than those of STY and STYR compounds to compensate for the oil content. Two compounds (STY and VIN) did not contain DCPD, while the other two (STYR and VINR) contained DCPD of 20 phr. Mixing was performed in a Banbury-type mixer, and the initial temperatures of the mixer were 110 and 80 °C for master batch (MB) and final mixing (FM) stages, respectively. The abrasion specimens were prepared by curing the compounds at 160 °C for the maximum cure time (t_max_) in a compression mold. The physical properties were measured using a Universal Testing Machine (Instron 6021, Norwood, MA, USA) at a cross-head speed of 200 mm/min.

### 2.2. Abrasion Tests

Four abrasion, namely CC, Lambourn, DIN, and LAT100 abrasion testers, were used. CC-2020 of Myungji Tech Co. (Seoul, Republic of Korea) was used as a CC abrasion tester, and the dimension of the specimen was an outer diameter of 50 mm, an inner diameter of 13 mm, and a thickness of 13 mm. The rotation speed of the CC sample was 750 rpm and the chipping speed was 60 rpm. The width of the chipping blade is 6 mm. The 2 specimens were tested for 10 min each. AB-1165 of Ueshima Seisakusho Co. (Osaka, Japan) was used as a Lambourn abrasion tester, and the dimension of the specimen was an outer diameter of 49 mm, an inner diameter of 23 mm, and a thickness of 10 mm. The speeds of the sample and abrasive wheel were 50 and 40 m min^−1^, respectively, with the slip ratio of 19.7%. The load was 44.8 N and the chamber temperature was 35 °C. The outer diameter of the abrasive wheel was 175 mm and the width was 25 mm. Then, 80-grit sandpaper was attached to the abrasive wheel. Talc was sprinkled between the specimen and abrasive with the minimal injection level. Each specimen was tested for 10 min.

WL210A of Withlab Co. (Gunpo-si, Republic of Korea) was used as a DIN abrasion tester. The dimension of the specimen was a diameter of 16 mm and a thickness of 8 mm. The diameter of the drum was 150 mm with 60-grit sandpaper and the rotation speed was 40 rpm. When the test was started, the specimen was moved 40 m from right to left of the tester on the rotating abrasive drum. The 2 specimens were abraded for 3 min each. The LAT100 tire tread compound tester (VMI group, Grlriaweg, The Netherlands) was used. The dimension of the specimen was a diameter of 80 mm diameter and a thickness of 19 mm. Electro Corundum Disc Grain 60 of VMI group (Grlriaweg, The Netherlands) was used as the abrasive disk. The load force was 75 N and the slip angle was 3°. The abrasion test was conducted for 1 h and the velocity was 25 km h^−1^.

After the abrasion test, the wear particles were collected and separated by size using a sieve shaker of Octagon 200 (Endecotts Co., Derbyshire, UK). Standard test sieves of 1000, 500, 212, 106, 63, and 38 μm were used. Morphologies of the wear particles were observed using a digital microscope (Leica DM4M, Leica Microsystems, Wetzlar, Germany) and an image analyzer (EGVM 35B, EG Tech. Co., Anyang-si, Republic of Korea). The abrasion specimen surface after the abrasion test was examined using the image analyzer.

### 2.3. Analyses of Bound Rubber Contents and Crosslink Densities

Bound rubber contents of the rubber compounds were determined by extracting the unbound materials such as free rubbers (unbound rubbers) and ingredients with toluene at room temperature for 6 days and with *n*-hexane at room temperature for 1 day. Then, they were dried at room temperature for 2 days. The weights of samples before and after the extraction were measured and the bound rubber contents (R_b_s) were calculated by Equation (1):R_b_ (%) = 100 × {W_fg_ − W_t_[m_f_/(m_f_ + m_r_)]}/{W_t_[m_r_/(m_f_ + m_r_)]}(1)
where W_fg_ is the weight of filler and gel, W_t_ is the weight of the sample, m_f_ is the fraction of the filler in the compound, and m_r_ is the fraction of the rubber in the compound.

Crosslink densities of the rubber vulcanizates were measured by the swelling method [24,51,52]. The sample size was 0.5 × 0.5 cm^2^ with a thickness of 3 mm. Organic additives in the sample were extracted using THF and *n*-hexane for 3 and 2 days, respectively, and the sample was thoroughly dried for 2 days at room temperature. The organic materials-extracted sample was soaked in toluene for 2 days at room temperature. The weights of organic materials-extracted and solvent-swollen samples were measured. The crosslink densities (*X_c_*s) were calculated using Flory–Rehner Equation (2) [53]:*X_c_* = −[ln(1 − ν_2_) + ν_2_ + χν_2_^2^]/[*V*_1_(ν_2_^1/3^ − ν_2_/2)](2)
where ν_2_ is the volume fraction of the crosslinked polymer, χ is the interaction parameter between the polymer and solvent, *V*_1_ is the molar volume of the swelling solvent. The ν_2_ was obtained by Equation (3):ν_2_ = (*m*_2_/ρ_2_)/[(*m*_2_/ρ_2_) + (*m*_1_/ρ_1_)](3)
where *m*_1_ and *m*_2_ are the solvent and specimen weights at equilibrium swelling, respectively, and ρ_1_ and ρ_2_ are the densities of the swelling solvent and unswollen rubber sample, respectively.

## 3. Results and Discussion

### 3.1. Bound Rubber Contents, Crosslink Densities, and Tensile Properties

Table 2 presents the bound rubber contents and crosslink densities of the samples. The bound rubber contents of STY samples (STY and STYR) were approximately 40% lower than those of VIN samples (VIN and VINR), even though SSBR1, used for the STY samples, has a three times higher molecular weight than SSBR2, used for the VIN samples. This discrepancy may be attributable to the higher 1,2-unit contents of SSBR2, known to interact better with silica than styrene and 1,4-unit [54,55]. The bound rubber contents of samples containing DCPD (STYR and VINR) were slightly lower than those of samples without DCPD (STY and VIN), indicating that DCPD prevents the formation of bound rubber. DCPD molecules can interact with the filler and prevent the contact between rubber and filler to reduce the bound rubber formation.

The crosslink densities of STY samples were higher than those of VIN samples, even though SSBR1 has lower 1,4-unit content (46 wt%) compared with SSBR2 (60 wt%), likely because of the higher molecular weight of SSBR1. In general, the crosslink density of an SBR compound is influenced by the molecular weight and 1,4-unit content [56]. The crosslink densities of samples containing DCPD were 6% and 10% lower than those of samples without DCPD for the STY and VIN samples, respectively, suggesting that DCPD inhibits sulfur-crosslinking reactions.

Table 3 summarizes the tensile properties of the four samples. Typically, the tensile properties of cured rubbers are related to their crosslink densities [57,58,59,60]. The moduli of samples followed the order of the crosslink density: STY > STYR > VIN > VINR. However, the orders of the elongation at break and tensile strength were the same or slightly different from those of the moduli and crosslink density: STYR > STY > VIN > VINR, with STY and STYR samples swapping orders for the crosslink density. This result indicates that the tensile strength is more influenced by the elongation at break than by the modulus. The elongation at break of the STY samples was greater than that of VIN samples, despite the STY samples having higher crosslink densities. This may be due to the higher molecular weight of SSBR1 than SSBR2.

### 3.2. Abrasion Rates

Table 4 outlines the abrasion rates obtained from the four abrasion tests. The abrasion rate of the VINR sample in the DIN abrasion test could not be obtained because the specimen dislodged from the jig in the initial stage. The orders of abrasion rates varied with the different abrasion testers. For the CC abrasion test, the order was STY > STYR > VIN >> VINR, while the orders for the Lambourn, DIN, and LAT100 abrasion tests were VIN > STY > VINR > STYR; STYR > STY > VIN; and VIN >> STY > VINR > STYR, respectively. The orders for the Lambourn and LAT100 abrasion tests were identical.

The order of the abrasion rate for the CC abrasion test was consistent with that of the crosslink density. This result indicates that a higher crosslink density can result in a higher abrasion rate in the CC abrasion test, attributable to the lower loss modulus associated with higher crosslink density, reducing the ability of the material to absorb shock from the blade impact. However, the order of the abrasion rate for the DIN abrasion test was opposite to that of the bound rubber content. This result implies that higher bound rubber content may lead to reduced abrasion in the DIN abrasion test. The abrasion rate orders for the Lambourn and LAT100 abrasion tests were not consistent with the orders of the bound rubber content, crosslink density, or tensile properties. The abrasion rates of samples containing DCPD (STYR and VINR) in the Lambourn and LAT100 abrasion tests were lower than those of samples without DCPD (STY and VIN), even though the STYR and VINR specimens had lower bound rubber contents and crosslink densities than the STY and VIN samples, respectively. This result indicates that DCPD enhances abrasion resistance in the Lambourn and LAT100 abrasion tests.

### 3.3. Size Distributions and Morphologies of Wear Particles

Figure 1 displays the size distributions of wear particles produced by the CC abrasion test. The size distributions of wear particles for the STY samples showed similar but different patterns compared with the VIN samples. For all samples, most of the wear particles were larger than 1000 μm, with a sharp decrease in distribution as the particle size decreased. However, no wear particles were sized less than 212 μm. The VIN samples had more than 1% wear particles sized 212–500 μm, whereas such particles were detected in trace amounts in the STY samples. The size distributions of wear particles of 500–1000 μm for the VIN samples were 4 and 29 times greater than those for the STY samples without and with the resin, respectively. The size distribution of wear particles > 1000 μm exhibited the following order: STYR (99.5%) > STY (93.6%) > VINR (85.6%) > VIN (74.0%), opposite to that of the bound rubber content. Given that the VIN samples had higher bound rubber contents and lower crosslink densities than the STY samples, it can be inferred that lower bound rubber content and higher crosslink density may result in the production of larger wear particles in the CC abrasion test. However, the size distributions of wear particles > 1000 μm for samples containing the resin were greater than those for the samples without the resin, regardless of the sample type, despite the crosslink densities of samples containing the resin being lower than those of samples without the resin. Hence, the influence of bound rubber content on the wear particle size distribution likely surpasses that of the crosslink density, causing the distribution to shift toward larger particle sizes as the bound rubber content decreases.

Figure 2 shows magnified images of the wear particles produced by the CC abrasion test. The wear particles exhibited irregular shapes with small aspect ratios. The VIN samples had rougher surfaces than the STY samples, and the samples containing the resin had slightly rougher surfaces than those without the resin. Assuming the filler part remains intact even when impacted by the blade, attributable to its solid structure surrounded by bound rubber, and the part of free rubber chains probably being broken by the external impact, these differences in the wear particle shapes can be explained. Higher bound rubdensity may also contribute to this roughness. The VIN samples had higher bound rubber contents and lower crosslink densities than those of the STY samples. Moreover, the samples containing the resin had lower crosslink densities than those without the resin, despite having lower bound rubber contents.

Figure 3 presents the size distributions of wear particles produced by the Lambourn abrasion test. The wear particles from the STY samples exhibited similar but distinct distribution patterns compared with the VIN samples. Most of the wear particles were larger than 1000 μm, with a steep decline in distribution observed as the particle size decreased, regardless of the sample types. For the wear particles > 1000 μm, the proportions in the VIN samples were higher in the Lambourn abrasion test than in the CC abrasion test, while the opposite was true for the STY samples. The Lambourn abrasion test generated wear particles across a wide range, from >1000 μm to 63–106 μm, with only trace amounts detected in the 63–106 μm range. Interestingly, the size distributions of wear particles sized 500–1000 μm (0.4–0.9%) were considerably lower than those of particles sized 106–212 μm (4.9–22.4%). For wear particles larger than 1000 μm, the samples containing the resin exhibited higher distributions than those without the resin. Conversely, for the wear particles < 500 μm, the distributions were lower in the samples with the resin. The STY samples (65.9% and 86.7%) had lower distributions of wear particles > 1000 μm than the VIN samples (88.3% and 93.4%), whereas the STY samples (33.7% and 12.4%) had higher distributions of wear particles < 500 μm than the VIN samples (11.1% and 6.0%). The size distribution order of wear particles > 1000 μm was VINR (93.4%) > VIN (88.3%) > STYR (86.7%) > STY (65.9%), opposite to that of the crosslink density. This result implies that the generation of wear particles > 1000 μm in the Lambourn abrasion test is influenced by the crosslink density. The addition of the resin decreases the crosslink density and increases slippage among rubber chains, leading to the production of more large wear particles > 1000 μm.

Figure 4 shows magnified images of the wear particles produced in the Lambourn abrasion test. Many talc particles were attached to the wear particle surface. Most of the wear particles > 1000 μm were long and noodle-like, while those < 1000 μm had stick-like shapes. Tire-road wear particles (TRWPs) found on real roads typically exhibit stick-like shapes. The wear particles generated in the Lambourn abrasion test had rough surfaces, unlike the smooth TRWPs typically found on real roads.

Figure 5 displays the variations in wear particle size distributions by sample type for the DIN abrasion test. The wear particle size distributions of the STY and VIN samples were significantly different and showed contrasting patterns: The distribution for the STY sample sharply decreased with decreasing particle size from >1000 μm to 500–1000 μm, whereas that of VIN sample significantly increased as the particle size decreased from >1000 μm to 500–1000 μm. The wear particles for the STY and STYR samples were mainly distributed in the >1000 μm range (91.1% and 95.1%, respectively), whereas the VIN sample exhibited most wear particles in the 212–500 μm (56.0%) range, followed by the 500–1000 μm range (36.1%). The size distribution of wear particles > 1000 μm for the VIN sample was only 7.2%. This may be due to the low elongation at break and tensile strength. In general, high friction is applied between the abrasion specimen and drum during the DIN abrasion test, causing rubber chains in the specimen with low elongation at break and tensile strength to break at low strain.

Figure 6 shows the magnified images of wear particles produced by the DIN abrasion test. These particles had more irregular structures and were rougher than those generated through the CC and Lambourn abrasion tests (Figure 2 and Figure 4, respectively). This is likely due to the high friction between the specimen and drum surface in the DIN abrasion test. No friction exists between the specimen and blade in the CC abrasion test, and the friction is reduced in the Lambourn abrasion test by the use of talc powder.

Figure 7 shows the variations in wear particle size distributions for different sample types based on the LAT100 abrasion test results. Except for the STYR sample, the wear particles were detected across a wide range of sizes from >1000 μm to 63–106 μm. The wear particle size distributions for the STY samples exhibited different patterns from those for the VIN samples. The STY and STYR samples exhibited the highest distributions at >1000 μm, with a significant decrease in distribution as the particle size decreased, while those of the VIN and VINR samples displayed nearly the same distributions at particle sizes of >1000 μm (42.3% and 37.5%, respectively) and 500–1000 μm (38.6% and 36.9%, respectively). For the wear particles > 1000 μm, the STY sample had a lower proportion than that of the STYR sample, whereas the VIN sample had a higher proportion than that of the VINR sample. The size distribution order of wear particles > 1000 μm was STYR (83.0%) > STY (68.4%) > VIN (42.3%) > VINR (37.5%), corresponding to the orders of elongation at break and tensile strength. This suggests that the size distribution of wear particles > 1000 μm generated from the LAT100 abrasion test is related to the elongation at break and tensile strength. Predominant productions of large wear particles > 500 μm from the abrasion testers are very different from the TRWPs on actual roads. In general, most TRWPs present on real roads are smaller than 200 μm, with large TRWPs > 500 μm rarely detected [14,15,16,17,18,19,61].

The wear particles produced by the LAT100 abrasion test had stick-like shapes but were more irregular and rougher compared with the TRWPs on real roads (Figure 8) [19,62]. Some abrasive wear particles were present on the wear particle surface, similar to TRWPs on real roads, though the quantity was considerably lower. No significant differences were observed across the different sample types.

### 3.4. Abrasion Patterns

The abrasion patterns on specimens after the abrasion tests were examined (Figure 9). The specimens after the CC abrasion test showed clear abrasion patterns with hollows, and the order of abrasion pattern spacing was STY > STYR > VIN > VINR, consistent with the order of the abrasion rate. The specimens from the Lambourn abrasion test had long-wave abrasion patterns, related to the generation of long, noodle-like wear particles > 1000 μm. The order of clearance of the abrasion patterns was VIN > STY > VINR > STYR, consistent with the order of the abrasion rate. The VINR specimen after the DIN abrasion test exhibited a severely torn pattern, and the order of surface roughness for the DIN specimens was STYR > STY > VIN, corresponding to the order of the abrasion rate. The specimens after the LAT100 abrasion test showed fine abrasion patterns, with the order of abrasion spacing being VIN > STY > VINR > STYR, consistent with the abrasion rate.

## 4. Conclusions

The orders of moduli and tensile strength were related to that of crosslink density. The orders of abrasion rates varied depending on the abrasion testers used: STY > STYR > VIN > VINR for the CC abrasion test, VIN > STY > VINR > STYR for the Lambourn abrasion test, STYR > STY > VIN for the DIN abrasion test, and VIN > STY > VINR > STYR for the LAT100 abrasion test. The orders of abrasion rates for the Lambourn and LAT100 abrasion tests were the same, the order for the CC abrasion test matched that of the crosslink density, and the order for the DIN abrasion test was the reverse of the order of the bound rubber content. The abrasion rates of the samples containing DCPD were lower than those of the samples without DCPD, except for the DIN abrasion test. The addition of the resin led to the generation of larger wear particles. The wear particles were most distributed at >1000 μm, except for the VIN sample and DIN abrasion test. The size distributions of wear particles > 1000 μm were 74.0–99.5%, 65.9–93.4%, 7.2–95.1%, and 37.5–83.0% for the CC, Lambourn, DIN, and LAT100 abrasion tests, respectively. The wear particles larger than 212, 63, 106, and 63 μm were produced in the CC, Lambourn, DIN, and LAT100 abrasion tests, respectively. The size distributions of wear particles from the STY and VIN samples generated through the DIN abrasion test showed opposite patterns, with the most frequent distribution of particle sizes for the VIN sample being 212–500 μm (56.0%). The wear particles produced in the CC abrasion test had irregular shapes with small aspect ratios. Talc particles were abundant on the wear particles produced in the Lambourn abrasion test, with most wear particles > 1000 μm exhibiting noodle-like morphologies, while those <1000 μm had stick-like shapes. The wear particles produced in the DIN abrasion test had more irregular structures and were rougher than those generated in the CC and Lambourn abrasion tests. The wear particles produced by the LAT100 abrasion test exhibited stick-like shapes but were more irregular and rougher compared with TRWPs on real roads. The order of abrasion pattern spacing corresponded to the order of the abrasion rate. The differences in abrasion behaviors could be explained by the bound rubber contents, crosslink densities, and tensile properties.

## Figures and Tables

**Figure 1 polymers-16-02038-f001:**
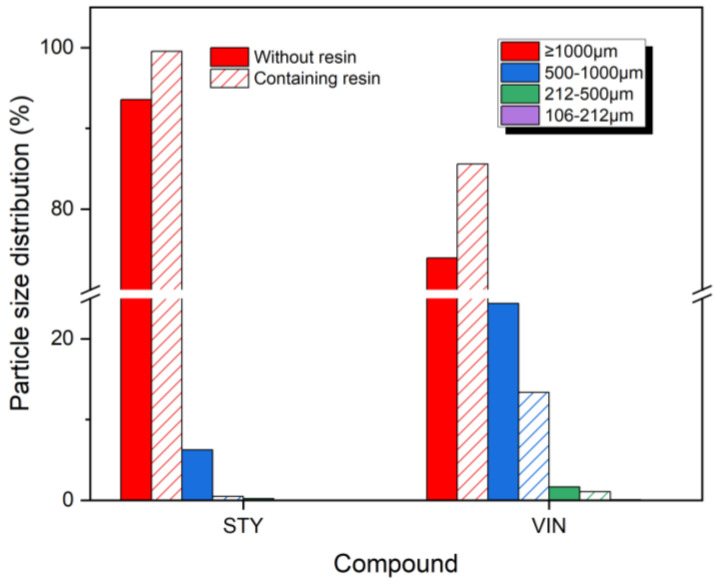
Size distributions of the wear particles produced by cut and chip (CC) abrasion tester. The solid and slash bars denote the samples without and containing the resin, respectively.

**Figure 2 polymers-16-02038-f002:**
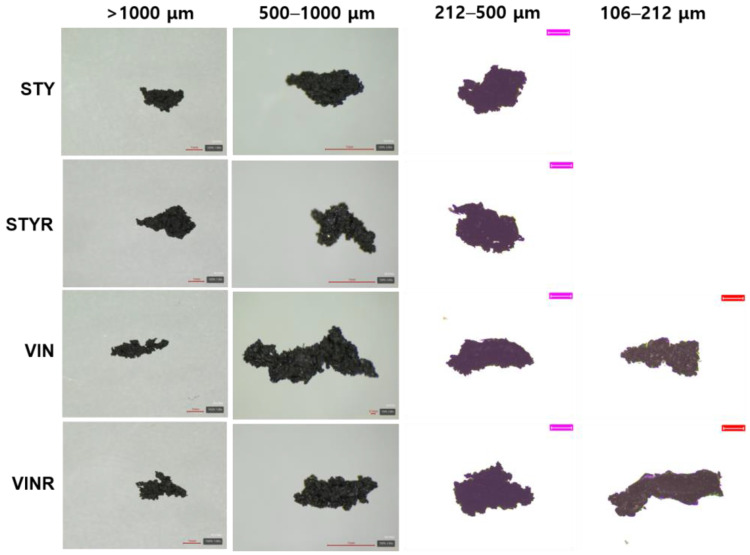
Magnified images of the wear particles produced by cut and chip (CC) abrasion tester. The scale bars of wear particles of >1000 μm are 1 mm, those of wear particles of 500–100 μm of STY, STYR, and VINR samples are 1 mm, that of the wear particle of 500–100 μm of the VIN sample is 0.1 mm, and the pink (212–500 μm) and red (106–212 μm) scale bars are 200 μm and 100 μm, respectively.

**Figure 3 polymers-16-02038-f003:**
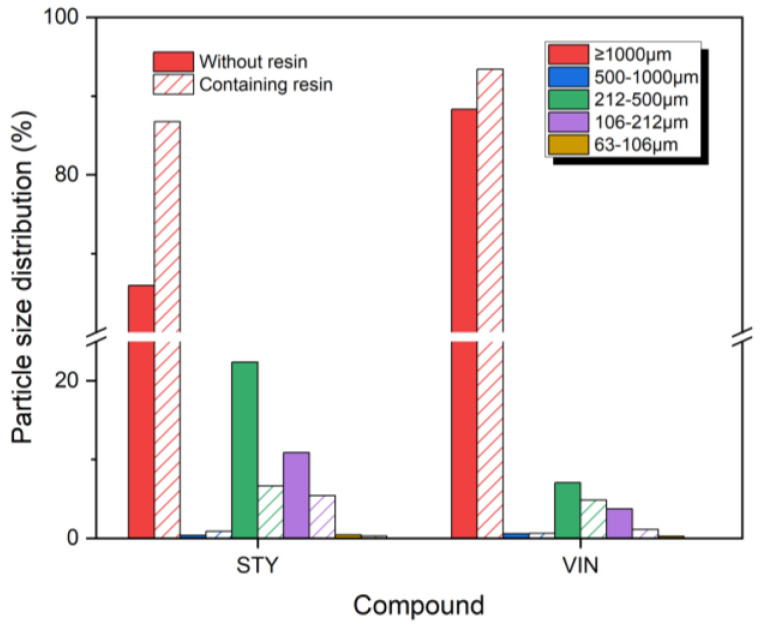
Size distributions of the wear particles produced by Lambourn abrasion tester. The solid and slash bars denote the samples without and containing the resin, respectively.

**Figure 4 polymers-16-02038-f004:**
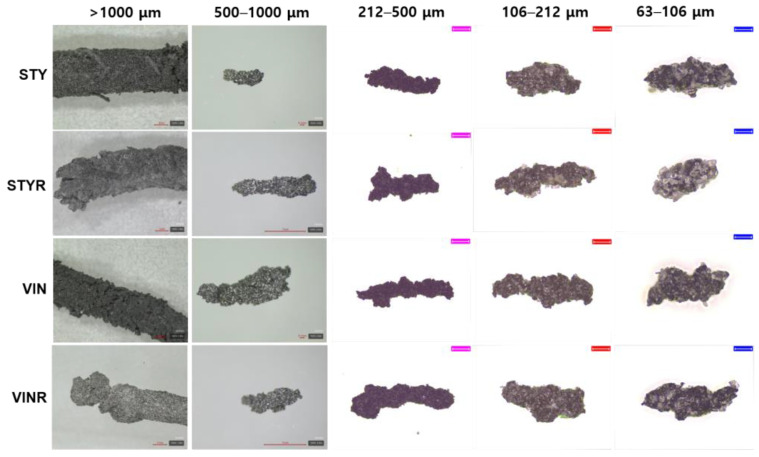
Magnified images of the wear particles produced by Lambourn abrasion tester. The scale bars of wear particles of >1000 μm are 1 mm, those of wear particles of 500–100 μm of the STY and VIN samples are 0.1 mm, those of wear particles of 500–100 μm of the STYR and VINR samples are 1 mm, and the pink (212–500 μm), red (106–212 μm), and blue (63–106 μm) scale bars are 200 μm, 100 μm, and 50 μm, respectively.

**Figure 5 polymers-16-02038-f005:**
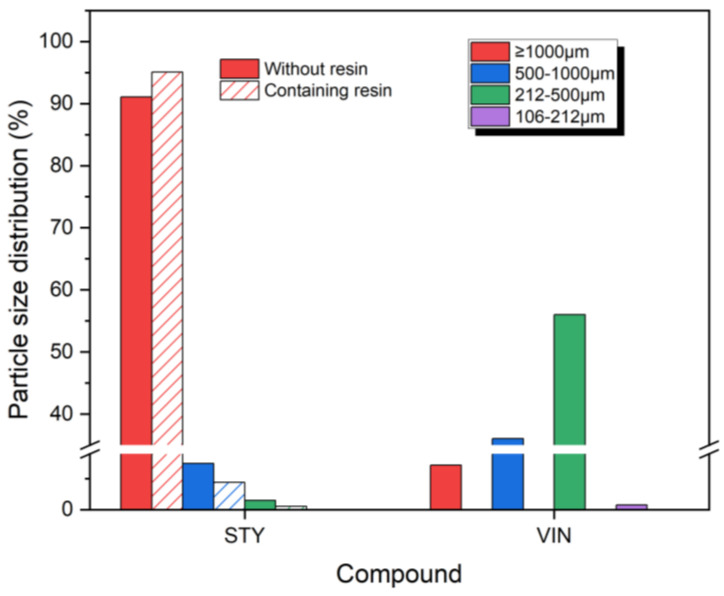
Size distributions of the wear particles produced by DIN abrasion tester. The solid and slash bars denote the samples without and containing the resin, respectively.

**Figure 6 polymers-16-02038-f006:**
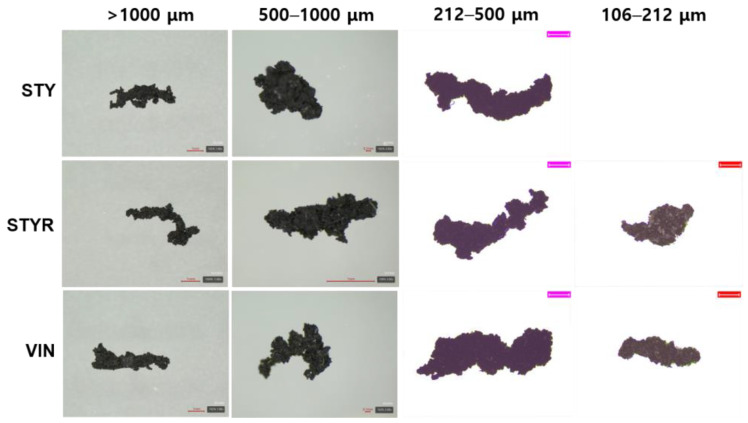
Magnified images of the wear particles produced by DIN abrasion tester. The scale bars of wear particles of >1000 μm are 1 mm, those of wear particles of 500–100 μm of the STY and VIN samples are 0.1 mm, that of wear particle of 500–100 μm of the STYR sample is 1 mm, and the pink (212–500 μm) and red (106–212 μm) scale bars are 200 μm and 100 μm, respectively.

**Figure 7 polymers-16-02038-f007:**
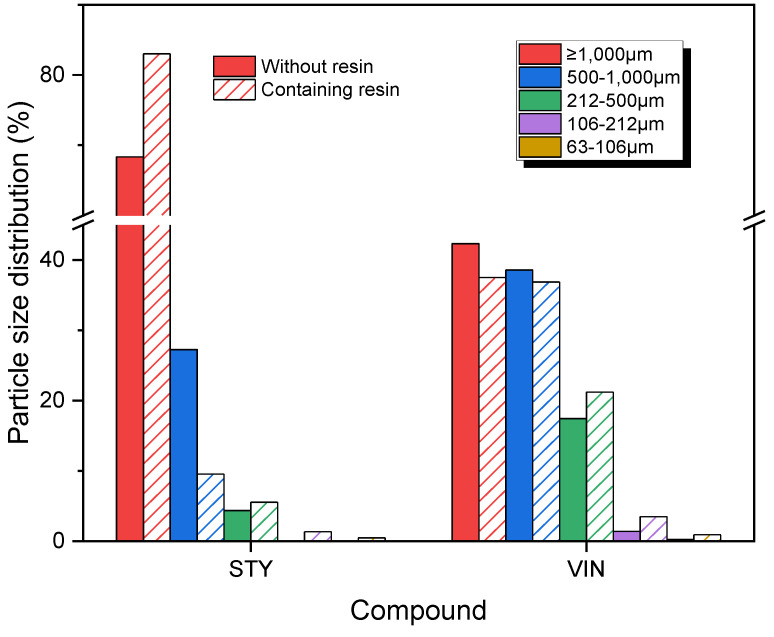
Size distributions of the wear particles produced by LAT100 abrasion tester. The solid and slash bars denote the samples without and containing the resin, respectively.

**Figure 8 polymers-16-02038-f008:**
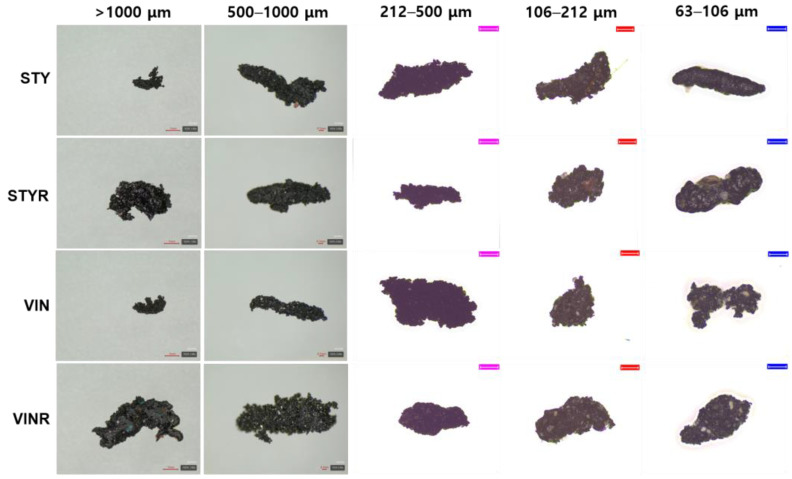
Magnified images of the wear particles produced by LAT100 abrasion tester. The scale bars of wear particles of >1000 μm and 500–100 μm are 1 mm and 0.1 mm, respectively, and the pink (212–500 μm), red (106–212 μm), and blue (63–106 μm) scale bars are 200 μm, 100 μm, and 50 μm, respectively.

**Figure 9 polymers-16-02038-f009:**
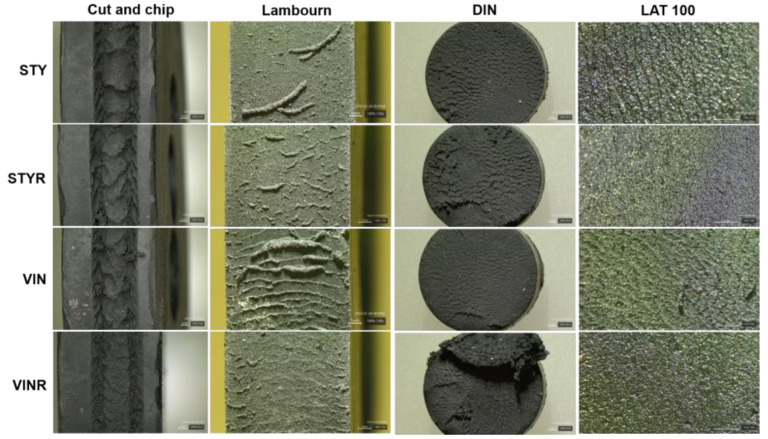
Magnified images of the abrasion specimens after the abrasion test. The scale bars were 1 mm.

**Table 1 polymers-16-02038-t001:** Formulation of the rubber compounds (phr) *.

Compound	STY	STYR	VIN	VINR
SSBR1	137.5	137.5	---	---
SSBR2	---	---	100.0	100.0
Silica (7000Gr)	80.0	80.0	80.0	80.0
Silane (X-50S)	8.0	8.0	8.0	8.0
DPG	1.6	1.6	1.6	1.6
Zinc oxide	4.0	4.0	4.0	4.0
Stearic acid	3.0	3.0	3.0	3.0
6PPD	2.0	2.0	2.0	2.0
TMQ	1.0	1.0	1.0	1.0
Wax	1.0	1.0	1.0	1.0
TDAE oil	5.0	5.0	42.5	42.5
DCPD resin	---	20.0	---	20.0
Sulfur	1.6	1.6	1.6	1.6
TBBS (NS)	1.1	1.1	1.1	1.1

* SSBR1 is solution SBR with high styrene content (styrene = 40 wt%, 1,2-unit = 14 wt%, and M_w_ = 1.65 × 10^6^–1.75 × 10^6^), and SSBR2 is solution SBR with high vinyl content (styrene = 15 wt%, 1,2-unit = 25 wt%, and M_w_ = 5.0 × 10^5^–6.0 × 10^5^). Acronyms: diphenylguanidine (DPG), N-(1,3-dimethylbutyl)-N′-phenyl-l,4-phenylenediamine (HPPD), polymerized 2,2,4-trimethyl-1,2-dihydroquinoline (TMQ), treated distillate aromatic extract (TDAE) oil, dicyclopentadiene (DCPD), and N-tert-butylbenzothiazole-2-sulfenamide (TBBS).

**Table 2 polymers-16-02038-t002:** Bound rubber contents and crosslink densities of the samples.

Compound	STY	STYR	VIN	VINR
Bound rubber content (%)	47.1	46.7	76.0	75.4
Crosslink density (10^−5^ mol cm^−3^)	6.63	6.22	5.41	4.85

**Table 3 polymers-16-02038-t003:** Tensile properties of the rubber compounds.

Physical Property	STY	STYR	VIN	VINR
100% modulus (MPa)	1.55	1.29	1.27	0.99
200% modulus (MPa)	3.16	2.55	2.34	1.79
300% modulus (MPa)	5.59	4.39	4.06	2.94
400% modulus (MPa)	8.65	6.74	6.54	4.62
500% modulus (MPa)	12.01	9.36	9.52	6.76
Elongation at break (%)	648	702	538	472
Tensile strength (MPa)	22.23	22.55	18.13	14.57

**Table 4 polymers-16-02038-t004:** Abrasion rates depending on the abrasion testers (mg min^−1^).

Compound	Abrasion Tester			
	Cut and Chip	Lambourn	DIN	LAT100
STY	92	59	106	7
STYR	78	51	182	3
VIN	70	77	61	69
VINR	15	52	---	4

## Data Availability

Data are contained within the article.

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
