# Peer review of "Abrasion Behaviors of Silica-Reinforced Solution Styrene–Butadiene Rubber Compounds Using Different Abrasion Testers"

_polymers, 2024, doi:10.3390/polym16142038_

Round 1
Reviewer 1 Report
Comments and Suggestions for Authors
- Line 153: Was the extraction carried out using toluene vapor? In a Soxhlet apparatus or by some other method?
- Line 184: Please add a discussion on why DCPD prevents the formation of bound rubber. It might already be described in the literature.
- Line 71: The experimental section explains the obtained results based on differences in the molecular weight of the rubbers, but the actual MW data is not provided in the Materials and Methods section.
- Line 299: The image quality for particles sized 63-106 m is quite low. For characterizing the surface of such small particles, it would be preferable to conduct the study using SEM.
- The labels (scale?) on Figures 4, 6, 8, and 9 are in very small font.
- The conclusions can be supplemented with data on mechanical properties and cross-link density.
Author Response
Q1. Line 153: Was the extraction carried out using toluene vapor? In a Soxhlet apparatus or by some other method?
A1. The organic ingredients in the sample was removed by solvent extraction at room temperature to prevent deformation of the sample.
Q2. Line 184: Please add a discussion on why DCPD prevents the formation of bound rubber. It might already be described in the literature.
A2. The sentence “DCPD molecules can interact with the filler and prevent the contact between rubber and filler to reduce the bound rubber formation.” was added in Lines 160-161.
Q3. Line 71: The experimental section explains the obtained results based on differences in the molecular weight of the rubbers, but the actual MW data is not provided in the Materials and Methods section.
A3. The molecular weights of two SSBRs were mentioned in the below of Table 1.
Q4. Line 299: The image quality for particles sized 63-106 mm is quite low. For characterizing the surface of such small particles, it would be preferable to conduct the study using SEM.
A4. The images for particles sized 63-106 mm look like a bit foggy. This is because of a lot of talc particles on the wear particle surface.
Q5. The labels (scale?) on Figures 4, 6, 8, and 9 are in very small font.
A5. The sizes of scale bars were marked in the figures and mentioned in the figure caption. There are many images of 12 – 20 images in one Figure, so the scale bars are relatively small but they can be distinguished.
Q6. The conclusions can be supplemented with data on mechanical properties and cross-link density.
A6. The sentence “The orders of moduli and tensile strength were related to that of crosslink density.” was added in Line 356. Relationship between abrasion rate and crosslink density was mentioned in Lines 361-362.
Reviewer 2 Report
Comments and Suggestions for Authors
The manuscript “Abrasion behaviors of silica reinforced solution styrene-butadiene rubber compounds using different abrasion testers refers to the properties of elastomeric materials used in tyre industry. I find this manuscript interesting and worth publishing because the data obtained by authors can be interesting for rubber industry. Authors used various test methods and different equipment to study the abrasion behaviour and compared the obtained results. Abrasion behaviour is very important in case of practical application of designed rubber materials.
However I find this manuscript interesting and worth publishing there are comments for authors which I would like to point before accepting that work. See added file

Author Response
Q1. Abstract. It is written concisely. Although could you reorganize abstract in more logical way, e.g start with the general description of the material used, the influence of the type of the SBR used and added resin on the abrasion rate, than describe the differences in the results when 4 various equipments were used during the abrasion tests.
A1. The general description of the materials and abrasion testers used in this study was mentioned in Lines 9-14. The influence of the SBR type was mentioned in Lines 17-19: “The abrasion rates of STY samples in the Lambourn and LAT100 abrasion tests were lower than those of VIN samples, whereas the values in the CC and DIN abrasion tests were higher than those of VIN samples. The wear particles were predominantly larger than 1000 mm, except for the VIN sample in the DIN abrasion test.”. The influence of the resin was mentioned in Lines 16-17: “The addition of the resin improved the abrasion rate and resulted in the generation of large wear particles.”.
Q2. Introduction. Could you more strongly underline the differences in these four methods. And add the comments about the advantages and disadvantages of every test equipment. Could authors in Introduction shortly underline the novelty of this research as compare with the cited literature, or add the short paragraph why the obtained results can be interesting for tyre industry?
A2. The sentences “There are a lot of various particles including wear particles of tire and road pavement and soils from the outside on real roads. Of the four abrasion tests, the Lambourn test uses mineral particles, however it may not apply various driving conditions.” were added in Lines 51-54.
Q3. Material and methods In this part should be added the information about the manufacturing company that synthetized elastomers used. Please add standards (ISO, ASTM, DIN e.g) according which the equipment used can operated. Please explain the acronyms used in Table 1, e.g DPG – diphenylguanidine , in Line 86-115 and in Line 120-149 is copied similar text, correct this.
A3. The manufacturers of SSBRs were not mentioned by request of the supplier. ASTM D 5963 for DIN abrasion test was added in Line 49. The full names of the acronyms were mentioned in the below of Table 1.
Q4. Results and discussion a) in Line 184 add the comment and explain more deeply why DCPD prevent the formation of bound rubber. b) In Line 198 explain more deeply why the elongation at break was lower for the samples with lower crosslink density, usually the elongation at break decrease as the crosslink density increase. The higher amount of bounded rubber could probably have impact on the mobility of the elastomeric chains and further on the elongation at break or not? Explain it. c) Explain more deeply why abrasion rates depended on the abrasion testers according to the differences in parameters and conditions of the used test methods (cheeping speed, speed of abrasion wheel, the rotation speed etc.) d) What about the influence of the changes in composition on curing parameters, the increase of torque during curing, the curing time etc., did authors study this?
A4. (a) The sentence “DCPD molecules can interact with the filler and prevent the contact between rubber and filler to reduce the bound rubber formation.” was added in Lines 160-161. (b) The result that the elongation at break of the STY samples were greater than those of VIN samples may be due to the higher molecular weight of SSBR1 than SSBR2. The sentence “This may be due to the higher molecular weight of SSBR1 than SSBR2.” was added in Lines 178-179. (c) In general, the abrasion rate is influenced by various parameters like the abrasive type, load, third body particles, cheeping speed, speed of abrasion wheel, rotation speed, and so on. The CC test results cannot be compared with the other results, because a CC tester uses a blade not abrasive. The Lambourn test results cannot be compared with the other results, because a Lambourn tester uses third body particles of talc. In order to compare the abrasion test results obtained from the different abrasion tests, a lot of detailed test results including the detailed various parameters are required. These subjects will be considered as future works. (d) We did not examine the influence of the changes in composition on curing parameters, the increase of torque during curing, the curing time etc. Abrasion behavior of a rubber vulcanizate should be influenced by the sample state. These subjects will be considered as future works.
Q5. Check again grammar in whole manuscript, especially the use of the, a.
A5. The manuscript was carefully checked, and the corrections were marked in blue.